# Mitochondrial genome of *Geomydoecus aurei*, a pocket-gopher louse

Theresa A. Spradling[‡]*, Alexandra C. Place, Ashley L. Campbell[¤], James W. Demastes[‡]

Department of Biology, University of Northern Iowa, Cedar Falls, Iowa, United States of America

¤ Current address: College of Osteopathic Medicine, Des Moines University, Des Moines, Iowa, United States of America
‡ TAS and JWD are joint senior authors.
* theresa.spradling@uni.edu

**Data Availability Statement:** Nanopore sequence data were accessioned accessioned at GenBank (PRJNA689128). Mitochondrial chromosome data were accessioned at GenBank (MW396892-396902).

## Abstract

Parasitic lice demonstrate an unusual array of mitochondrial genome architectures and gene arrangements. We characterized the mitochondrial genome of *Geomydoecus aurei*, a chewing louse (Phthiraptera: Trichodectidae) found on pocket gophers (Rodentia: Geomyidae) using reads from both Illumina and Oxford Nanopore sequencing coupled with PCR, cloning, and Sanger sequencing to verify structure and arrangement for each chromosome. The genome consisted of 12 circular mitochondrial chromosomes ranging in size from 1,318 to 2,088 nucleotides (nt). Total genome size was 19,015 nt. All 37 genes typical of metazoans (2 rRNA genes, 22 tRNA genes, and 13 protein-coding genes) were present. An average of 26% of each chromosome was composed of non-gene sequences. Within the non-gene region of each chromosome, there was a 79-nt nucleotide sequence that was identical among chromosomes and a conserved sequence with secondary structure that was always followed by a poly-T region. We hypothesize that these regions may be important in the initiation of transcription and DNA replication, respectively. The *G. aurei* genome shares 8 derived gene clusters with other chewing lice of mammals, but in *G. aurei*, genes on several chromosomes are not contiguous.

## Introduction

Parasitic lice (Order Phthiraptera) have mitochondrial (mt) genomes that are exceptionally variable among species and strikingly atypical of the genomes observed in most other insects or other bilaterian animals. Whereas most bilaterian animals examined have a single, large mitochondrial chromosome [1, 2], the mt genome of some parasitic lice occurs as multiple smaller chromosomes. The initial discovery of an atypical genome was made in 2009 when Shao et al. characterized 18 small, circular chromosomes of the mt genome of the human louse (*Pediculus humanus;* [3]). That observation has been followed by numerous other examples of atypical mitochondrial genome architecture in other parasitic lice, with species exhibiting anywhere from 2–20 chromosomes [4–16].

**Funding:** Portions of this work were funded by National Science Foundation Grant DEB 1445708 to TAS and JWD. The Galaxy server that was used for some calculations is in part funded by Collaborative Research Centre 992 Medical Epigenetics (DFG grant SFB 992/1 2012) and German Federal Ministry of Education and Research (BMBF grants 031 A538A/A538C RBC, 031L0101B/031L0101C de.NBI-epi, 031L0106 de. STAIR (de.NBI)). The funders had no role in study design, data collection and analysis, decision to publish, or preparation of the manuscript.

**Competing interests:** The authors have declared that no competing interests exist.

Among parasitic lice, sucking lice (suborder Anopulura), the lice of elephants and warthogs (Suborder Rhynchophthirina), and the chewing lice of mammals (Family Trichodectidae; [13]), form a monophyletic assemblage [13]. All members of this assemblage studied to date exhibit highly fragmented mt genomes [13]. The exceptionally simple mt-genome architecture of this group is characterized by small chromosomes with only one to two protein-coding genes or one rRNA gene per chromosome along with one, two, three, or four tRNA genes and a non-coding region that bears motifs shared among chromosomes within each species [5–16].

In this study, we characterized the mitochondrial genome of *Geomydoecus aurei*, a chewing louse (Phthiraptera: Trichodectidae) found on pocket gophers (Rodentia: Geomyidae). One mitochondrial chromosome bearing the cytochrome oxidase *c* subunit I gene (*cox1*) and the *trnI* (isoleucine) gene of this species was previously characterized [17]. Herein, we characterize 11 additional chromosomes bearing the remaining 35 genes expected for an insect. We used a combination of methods including sequence reads from both Illumina and Oxford Nanopore sequencing coupled with wet-lab methods involving PCR, cloning, and Sanger sequencing to verify structure, sequence, and arrangement.

Pocket gophers and their chewing lice, including *G. aurei*, have been the subjects of considerable study with respect to cophylogeny (e.g., [18–20] and references therein). More recently, *G. aurei* has been the subject of a long-term ecological and population genetics study that spanned the course of 25 years [21, 22]. Beyond cataloging the chromosomal architecture of another species with an atypical genome, providing a better understanding of the genetics of chewing lice that inhabit pocket gophers will provide a foundation to launch more detailed studies of cospeciation and population genetics using mitochondrial sequences.

## Methods

### Louse collection

Pocket gophers (*Thomomys bottae*) were collected from Socorro County, New Mexico, with the approval from the New Mexico Department of Game and Fish (Permit #3500). Animals were collected using procedures in keeping with guidelines set by the American Society of Mammalogists [23] as described by Pietan et al. [17]. This study was approved by the University of Northern Iowa Institutional Animal Care and Use Committee. Because *G. aurei* individuals are small, approximately 0.2 mg, multiple lice were needed for this analysis. Lice used in this study were collected from the same host species collected in the same county, with a maximum geographic distance of 28 km between specimens.

### *rrnL* (*16s*) chromosome sequencing using universal primers

To characterize the mitochondrial chromosome bearing the *rrnL* gene, DNA was extracted from the same *G. aurei* specimen used by Pietan et al. [17] to characterize the *cox1* chromosome (JWD 39.6; New Mexico: Socorro Co.; 3.5 mi. S La Joya, 34.317, -106.857; GenBank: KX228450.1). Universal *rrnL* polymerase chain reaction (PCR) primers Lx16SR and 16SF (Table 1) were used [8, 12], with 39 amplification cycles and an annealing temperature of 45˚C. Amplification produced a DNA product approximately 300 nucleotides (nt) in length. This PCR product was treated with ExoSap-IT (USB, Cleveland, OH) and sequenced directly at the Iowa State University DNA Facility using the amplification primers. The sequenced product was used to generate primers 16S213F and 16S162R (Table 1), which were designed to amplify outward from each other to determine the remaining portion of the *rrnL* chromosome. Because cloning was to be used to allow sequencing through regions of heteroplasmy, a high-fidelity amplification mixture (Platinum SuperFi Green PCR Master Mix; Invitrogen, Carlsbad, CA) was used for this reaction. Thermal cycles were: 2 minutes at 95˚C, then 39

**Table 1. Primers (5'→3') used to generate mitochondrial sequences of *Geomydoecus aurei*.**

| Chromosome | Primer | Sequence (5'-3') |
|---|---|---|
| *rrnL* | Lx16SR | GACTGTGCTAAGGTAGCATAAT |
| | 16SF | TTAATTCAACATCGAGGTCGCAA |
| | 16S213F | CAGGGTCTTCTCGTCCAGTC |
| | 16S162R | GAAGGGGGAGCCAAACAGTT |
| Multiple circles | Multicircle 1 | TTCTATTTTGCACCTTTTGAACGG |
| | Multicircle 3 | CTTGTAGTTGTGAACTATTAGG |
| *atp6* | ATP6587F | TCCCACCCGTGATCGATAGT |
| | ATP6240R | CCGGCCTATATCACTAGCCG |
| *cox2* | 20Fcox2 | TTGTGGTGCTCTACATAGATTT |
| | 25Rcox2 | CCACAAATCTCTGAACACTGACC |
| *cox3* | cox3710F | CTCTTCGGTATGAGCGCCTT |
| | cox3500R | ACCCATGGTGTGCTCATGTT |
| | cox3444F | AACCAATAACCCCAGCCGAG |
| | cox3974R | CGAGAGGCCTGCAGTTTGTA |
| *cob* | cytb494F | ACACCACCACATATCCAGCC |
| | cytb407R | CTACCGCTAGGCCAACCAAA |
| *rrnS* | 12S294F | CCCTTTCGCTATGCCCTTCA |
| | 12S214R | CTTCAATTGGCACACACGCA |
| *nad1* | 374Fnad1 | GTCAAACCGGATTCGTGGGA |
| | 366Rnad1 | TGTCCACTGGTTGGGTTGTC |
| *nad3* | ND3DashF4 | CCCAGTCTCTGCAAGTAAA |
| | ND3DashR3 | GAGATTAACCGTATCCCATTTC |
| | ND3DashF6 | GAAGAATACTTCCCACTACC |
| | ND3DashR6 | GGAGAGAAGTCTAGGGTTTG |
| *nad4* | 310Fnad4 | ACATCTCAGGGATCGGTTAGTC |
| | 179Rnad4 | ACTTCCATGATTTTGGGTTGGA |
| *nad 5* | ND5DashR5 | GTGGAGTCTGAGAACTTAATC |
| | ND5DashR3 | CCTCCACTGAGAATCTCTAC |
| | ND5DashF5 | CTTTGGTCATATCCACTTATCC |
| | ND5DashF2 | GAAGGCCTACTTTCCTTTG |
| | ND5DashR7 | CTCAGATATCCAAAGGCTAAC |

cycles of 45 seconds at 94˚C, 45 seconds at 45˚C, 45 seconds at 72˚C, followed by 10 minutes at 72˚C. This amplification yielded a product that was approximately 1,500 nt in length. The product was cloned using the Zero Blunt TOPO 4 PCR cloning kit according to the manufacturer's recommendations (Invitrogen, Carlsbad, CA), and clone DNA was reamplified and sequenced with primers 16S213F and 16S162R as described above. Sequences were edited manually, and consensus sequences of forward and reverse reactions were created and aligned to each other using Geneious 11.0.4 [24] to generate a full *rrnL* chromosome sequence with double sequencing coverage.

## Cloning and sequencing by targeting a common sequence

Comparison of DNA sequences of the *G. aurei cox1* and the *rrnL* chromosomes revealed a 79-nt region of identical nucleotide sequence. Hypothesizing that this common sequence may be found on additional chromosomes, primers were developed to target this region with an orientation to amplify outward from the 79-nt sequence (primers Multicircle 1 and Multicircle

3; Table 1). For this analysis, additional DNA was isolated as described above from three *G. aurei* lice collected 12 km away from those used to determine the *cox1* and *rrnL* chromosomes (specimens TAS 845.33, 845.35, and 845.37; New Mexico: Socorro Co.; vicinity of Polvadera, 34.22008, -106.90454). Because these primers generated a heterogeneous mixture of amplification products, a high-fidelity amplification mixture (Platinum SuperFi Green PCR Master Mix; Invitrogen, Carlsbad, CA) was used to generate PCR products that were then cloned as described above. Clone DNA was then amplified and sequenced as above using plasmid primers. Sequences were edited manually, and consensus sequences of forward and reverse reactions were created and aligned using Geneious 11.0.4 [24]. Resulting contigs were subjected to BLASTn searches [25] against the National Center for Biotechnology Information's *nr/nt* database (https://blast.ncbi.nlm.nih.gov/). This strategy produced nucleotide sequences for *atp6/ atp8*, *cob*, and *rrnS*, which were used to generate additional primers for amplification and sequencing of the remainder of the mitochondrial circle (Table 1), resulting in 2-4x sequencing coverage of each chromosome. This strategy also produced nucleotide sequences for *nad2* that were combined with sequences from genomic analysis (below).

## Genome analysis

Illumina sequencing of DNA extracted from fifteen *G. aurei* taken from a single host individual (TAS 750; New Mexico: Socorro Co., 1.4 mi S, 0.8 mi W Las Nutrias, 34.45405556, -106.78277778) was performed as described by Light et al. [26]. These reads were trimmed using SolexaQA [27] and assembled in Geneious Version 11.04 [24]. Fifty percent of the paired-end reads were used and the minimum length for contigs was set at 131. Resulting contigs were used as a local BLAST database within Geneious [24] and suitable sequences were used for the production of primers for *cox2/nad4L*, *cox3*, *nad1*, *nad4*, and *nad5* chromosomes (Table 1), which were designed to amplify outward from these genes to cover the remainder of the chromosome. Resulting sequence data were aligned with the original contig data and used to generate additional sequencing primers (Table 1) where necessary to generate 2–4× coverage for each chromosome. Illumina sequences also were used in combination with cloned sequences (above) to reconstruct a *nad2* chromosome with 2x coverage.

An additional genomic analysis was performed using an Oxford Nanopore GridION X5 (Oxford Nanopore, Oxford, UK) at the Iowa State University DNA Facility. For this analysis, high molecular weight DNA was extracted from a pool of 132 frozen lice (ca. 25 mg) using the Nanobind Insect Big DNA Kit (Circulomics, Baltimore, MD). These lice came from a single host individual (TAS 866; New Mexico: Socorro Co., vicinity of Polvadera, 34.20483, -106.89633), and sequence read data were accessioned at GenBank (BioProject PRJNA689128). The Oxford Nanopore GRIDIon instrument produced 7,049 megabases (mb) of sequence data with an N50 of 13866. Currently, two annotated genomes are available for phthirapterans: *Pediculus humanus* [28] and *Columbicula columbae* [16] with genome sizes of 108 (mb) and 208mb respectively. Assuming *G. aurei* approximates a similar genome size, the GridION analysis achieved 30x - 60x coverage of the genome. The CANU program [29] was implemented via the Europe Galaxy Instance (UseGalaxy.eu) to correct, trim and assemble the sequences produced by the Nanopore GridION analysis. Two local BLAST databases were constructed: one of corrected and trimmed reads, and one with corrected, trimmed and assembled contigs. These databases were subjected to BLAST analysis and used to design outward-facing primers for the *nad5* and *nad6/nad3* chromosomes (Table 1) as needed to yield 2-4x coverage for each chromosome.

Our attempts at chromosome assembly from Nanopore and Illumina data were fraught with several difficulties including sequences in the genome that were common to multiple

chromosomes, extensive heteroplasmy in portions of each chromosome, inherent weaknesses in the data sets, and limited experience on our part. The relative contribution of each of these factors to the difficulty in assembling chromosomal-length sequences from next generation data is not clear to us. In the case of the Illumina data set, mitochondrial sequences of *G. aurei* seemed to be low in copy number and included significant confounding sequence from another louse species included in the same Illumina reaction. Chromosomal-length Illumina contigs were created for *nad2*, but not for other chromosomes. Nanopore data contained multiple disagreements in what appeared to be common sequences, contributing to difficulties in alignment. Regardless, because we were unable to create chromosome-scale assemblies, we found it more productive and reliable to use the genomic data as a foothold to allow access to individual chromosomes through standard PCR, cloning, and sequencing.

### Annotation and analysis of genome size

The online software program MITOS2 was used to determine locations of tRNAs, origin of replication regions, and protein-coding and rRNA genes on each chromosome [30, 31]. The program determines protein-coding genes by detecting similarity in the results of BLASTx searches against the amino acid sequences of the annotated proteins of metazoan mitochondrial genomes. The tRNA genes are annotated using a structure-based covariance model, and annotations for rRNA genes are established using structure-based covariance number models similar to the tRNA models [31]. Finally, Mitos2 uses nucleotide sequences from RefSeq and employs the protein prediction pipeline of MITOS using BLASTN to identify origins of replication on the heavy strand, and it uses a covariance model built from a manually curated set of all light-strand origins of replication annotated in RefSeq 65 (Alexander Donath, personal communication). To assess the beginning and end of protein-coding and rRNA genes that had been determined through the Blast analysis of Mitos2, alignments of nucleotide and amino acid sequences were created with *Trichodectes canis*, the most closely related trichodectid louse available [13], and manually adjusted in Geneious 11.0.4 [24]. Web-Beagle [32] was used to align rRNA sequences against *T. canis*, to examine secondary structure, and to compare regions of structural similarity.

A second computer program, ARAGORN [33], was employed using the Europe Galaxy Instance (UseGalaxy.eu; [34]) to locate and characterize potential tRNA sequences on each minicircle. ARAGORN incorporates searches for subsets of B-box consensus sequences followed by the construction of the associated T-loop and T-stem. If this process locates a B-box sequence, a search is then conducted for a subset of A-box consensus sequences, which is then followed by construction of the tRNA D-loop.

We applied linear regression analysis (Excel © 2017, Microsoft) to chromosome size vs. gene(s) size per chromosome. We also used linear regression to compare chromosome size and non-gene size per chromosome and to compare gene size vs. non-gene size. Geneious 11.0.4 [24] was used to search for direct repeats in the nucleotide sequences of chromosomes. Annotated chromosome reconstructions were accessioned at GenBank (Table 2).

## Results

### Chromosomal architecture

The 11 circular mitochondrial chromosomes characterized in this work, together with the one characterized by Pietan et al. [17], ranged in size from 1,318 nt to 2,088 nt and carried all 37 genes typical of metazoans (2 rRNA genes, 22 tRNA genes, and 13 protein-coding genes [1, 2]; Table 2). Total genome size was 19,015 nt. Mean chromosome size was 1,585 nt (standard deviation ± 252 nt). Each of the chromosomes of the *G. aurei* genome included only one or

**Table 2. *Geomydoecus aurei* mitochondrial chromosomes, named for their major protein-coding or rRNA genes.**

| Chromosome Name | Chromosome Gene Order [a] | Size (nt) of chromosome | Size (nt) of protein/rRNA Genes | Non-gene Region Size (nt) [b] / % of chromosome | tRNA Genes with Anti-codon |
|---|---|---|---|---|---|
| *rrnS* MW396892 | *-L1-rrnS* | 1,318 | 700 | 453 in 2 segments / 34% | *trnL1* (tag) |
| *rrnL* MW396893 | *H-rrnL-V* | 1,545 | 1,185 | 355 in 1 segment / 23% | *trnH* (gtg), *trnV* (tac) |
| *atp8/6* MW396902 | *L2-atp8-atp6* | 1,331 | 162 (*atp8*), 678 (*atp6*) | 338 in 2 segments / 25% | *trnL2* (gaa) |
| *cob* MW396901 | *E-cob-S1* | 1,545 | 1,086 | 332 in 1 segment / 21% | *trnE* (ttc), *trnS1* (tct) |
| *cox1* [c] KX228450 | *I-cox1* | 1,914 | 1,536 | 315 in 1 segment / 16% | *trnI* (gat) |
| *cox2-nad4L* MW396900 | *Y-cox2-R-nad4L* | 1,511 | 627 (*cox2*), 279 (*nad4L*) | 478 in 2 segments / 32% | *trnY* (gta), *trnR* (tcg) |
| *cox3* MW396899 | *P-cox3* | 1,356 | 786 | 504 in 2 segments / 37% | *trnP* (tgg) |
| *nad1* MW396898 | *W-nad1-Q* | 1,463 | 918 | 337 in 2 segments / 23% | *trnW* (tca), *trnQ* (ttg) |
| *nad2* MW396897 | *N-nad2-C* | 1,519 | 981 | 415 in 2 segments / 27% | *trnN* (gtt), *trnC* (gca) |
| *nad6/3* MW396896 | *F-nad6-M-T-G-nad3* | 1,503 | 441 (*nad6*), 327 (*nad3*), | 473 in 6 segments / 31% | *trnF* (gaa), *trnM* (cat), *trnT* (tgt), *trnG* (tcc) |
| *nad4* MW396895 | *K-nad4-A-S2* | 1,922 | 1,338 | 404 in 2 segments / 21% | *trnK* (ttt), *trnA* (tcg), *trnS2* (tga) |
| *nad5* MW396894 | *D-nad5* | 2,088 | 1,596 | 425 in 1 segment / 20% | *trnD* (gtc) |

All 37 genes typical of animal mtDNA are represented. Chromosome name with Genbank Accession number, gene order, chromosome size, gene size, non-gene content, and tRNA composition with anti-codons are given. DNA lengths are given in nucleotides (nt).

[a] tRNA genes are listed by their single letter code.

[b] excludes protein-coding, rRNA, and tRNA genes that are presumed functional

[c] [17]

two protein-coding genes or one rRNA gene. Each chromosome also included one, two, three, or four tRNA genes.

Chromosomes were compact, with the majority of each chromosome devoted to genes. Chromosome size and gene size showed a significant relationship, with larger gene content strongly associated with larger chromosomes ($F = 74.1$, $p < 0.0001$, $R^2 = 0.88$). However, there was no significant relationship observed between chromosome size and non-gene content. Likewise, gene size and non-gene content did not exhibit a significant relationship. Non-gene regions contributed an average of 26% to the length of each chromosome (Table 2). Gaps between genes were prevalent (Fig 1). Seven of the 12 *G. aurei* chromosomes exhibited multiple non-gene regions interspersed among the genes (Fig 1, Table 2).

## Motifs shared among chromosomes

Every chromosome had a 79-nt region that was identical on all chromosomes and that ran in a direction that was consistent among chromosomes relative to the direction of the genes (Figs 1 and 2). This 79-nt sequence is non-repetitive and starts with a high (70%) AT content that suddenly transitions to a high (58%) GC content 19 nt before the end of the sequence.

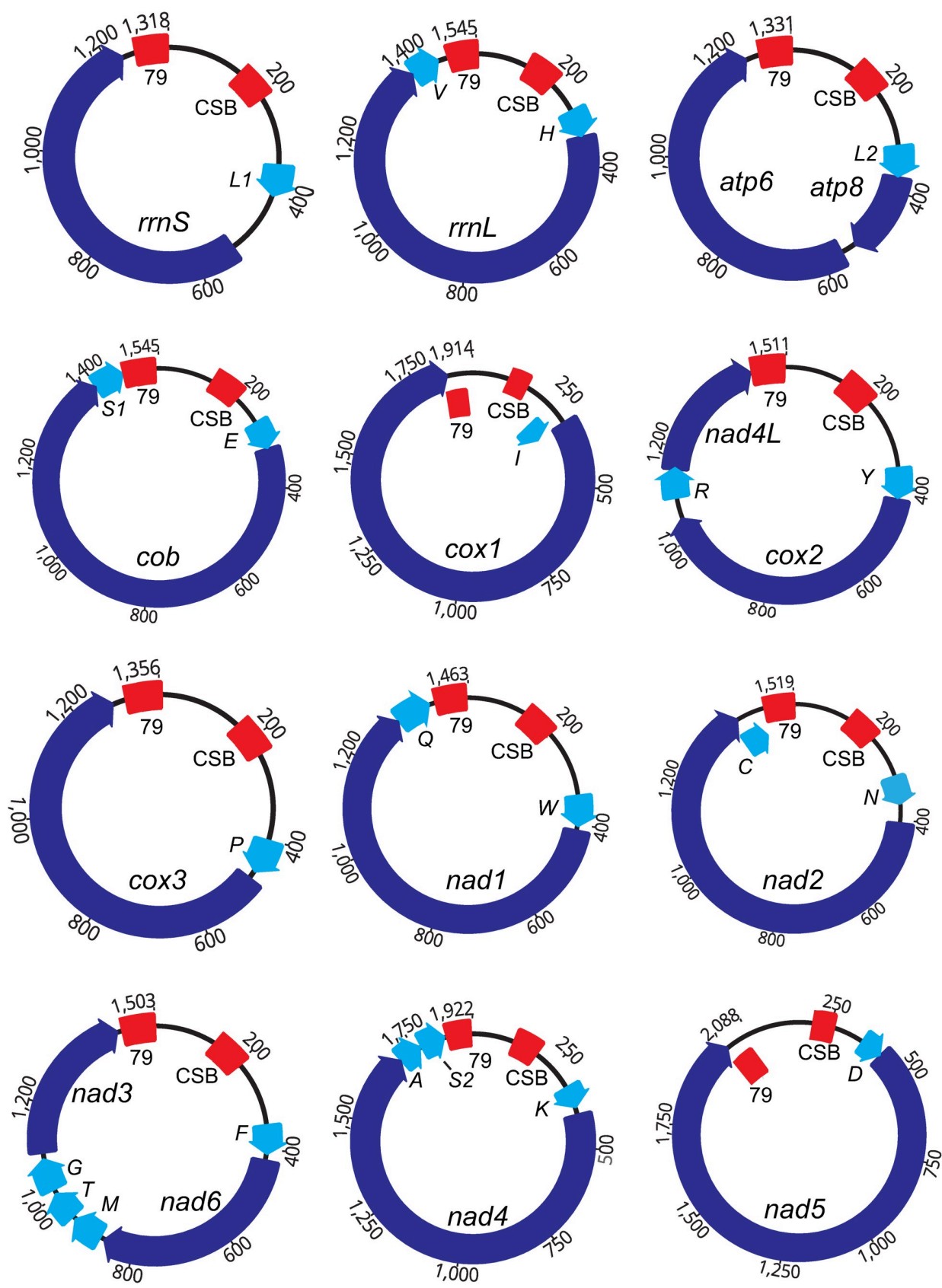

**Fig 1. Mitochondrial genome organization of *G. aurei*.** Name and orientation are given for each gene, indicated by arrows. The largest number on the outside of each circles indicates its total length (nt). Non-gene regions are shown as thin lines. Sizes are proportional. Gene names are: *atp8* and *atp6* (for ATP synthase subunits 8 and 6), *cox1-3* (for cytochrome *c* oxidase subunits 1–3), *cob* (for cytochrome *b*), *nad1-6* and *nad4L* (for NADH dehydrogenase subunits 1–6 and 4L), *rrnS* and *rrnL* (for small and large subunits of ribosomal RNA). tRNA genes are designated by the single-letter abbreviations of their corresponding amino acids. "79" refers to a 79-nt sequence found on all chromosomes. "CSB" (Conserved Sequence Block) indicates a DNA sequence that is highly similar among chromosomes (Fig 2).

In addition to the 79-nt common sequence, all chromosomes had a block of sequence that was highly similar among them (a "Conserved Sequence Block" or CSB; Fig 2). These sequences were identified as potentially functional copies of the *trnF* gene on some chromosomes, an indication of secondary structure. The CSB sequences were located 108–205 nt downstream of the 79-nt common sequence (average of 128 nt; Fig 1). Within this intervening space between the 79-nt common sequence and the CSB, MITOS [31] identified a putative origin of replication near the *trnF* sequence in 11 of the chromosomes (the software did not recognize an origin of replication at all in the *nad4* chromosome). The CSB exhibited direct repeat regions in the sequence for all chromosomes except the *nad5* chromosome, and it was followed by a poly-thymine stretch of 9–11 nucleotides that occurred 11–32 nucleotides downstream from the start of the CSB (distance was 11 nucleotides in 1 chromosome, 12 nucleotides in 3 chromosomes, 31 nucleotides in 1 chromosome, and 32 nucleotides in 7 chromosomes). Shifts in AT composition were also noted near the CSB, with high AT content (roughly 75% AT) upstream and downstream of the CSB and low AT content (44–53% AT) within the CSB.

## Primitive vs. derived gene arrangements

Song et al. [13] reported eight derived gene clusters common to parasitic lice (Phthiraptera): *trnI-cox1*, *trnY-cox2*, *trnE-cob-trnS1*, *nad1-trnQ*, *trnG-nad3*, *trnR-nad4L*, *trnK-nad4*, and *rrnL-trnV*. All eight of the derived gene clusters observed in most trichodectid and Anopluran lice were also observed in *G. aurei* (Fig 1, Table 2). However, despite these similarities, chromosome arrangement in *G. aurei* was unique among the trichodectid species reported to date [13], differing in at least three important aspects. In particular, while the *nad4L* and *cox2* genes are collocated on the same chromosome in *G. aurei*, they are not adjacent to each other as they are in other trichodectid species examined. Likewise, while *atp8* and *atp6* are collocated on the same chromosome, they are not adjacent to each other in *G. aurei* as they are in other trichodectid species examined. Finally, while *nad3* and *nad6* are collocated on the same chromosome, they are not adjacent to each other in *G. aurei* as they are in the trichodectid species, *T. canis*.

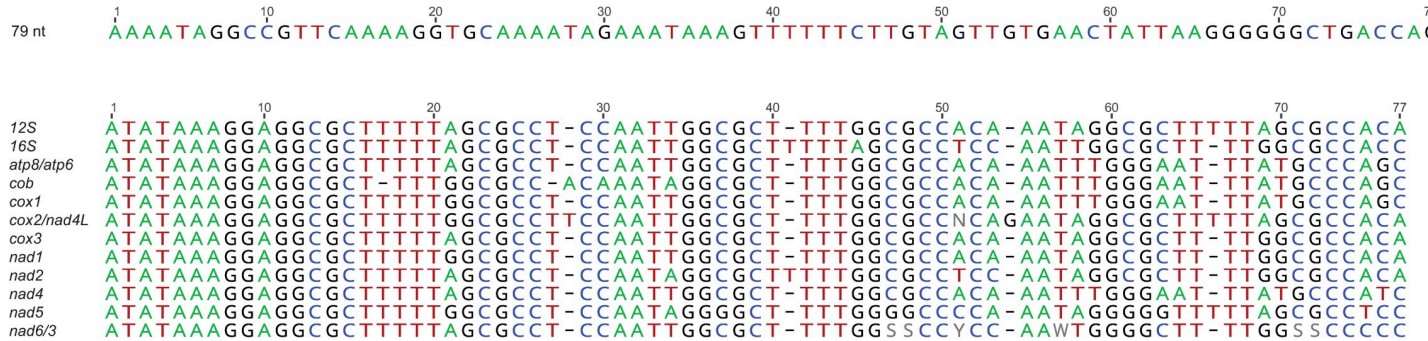

**Fig 2. Sequence of the 79-nt sequence common to all *Geomydoecus aurei* mtDNA chromosomes and alignment of a 77-nt sequence that is found 108–205 nt downstream, which is referred to here as the "CSB" (Conserved Sequence Block).** Some of these sequences were identified as having secondary structure similar to that of a tRNA gene in some analyses.

## Discussion

### Chromosomal architecture

Parasitic lice exhibit an impressive diversity of gene arrangement and chromosome number in their chromosomal architectures. To date, with the exception of a few very closely related species pairs, all species have a unique chromosome content and gene arrangement [14]. The *G. aurei* mitochondrial genome analyzed herein also was unique, though chromosome gene content bears most similarity to the genome of *T. canis*, the louse found on domestic dogs and wild canids. While the mitochondrial genome of *T. canis* includes 13 chromosomes, all 37 genes of the *G. aurei* genome appear on 12 chromosomes. The smaller number of chromosomes in *G. aurei* is attributable to the lack of a tRNA-only chromosome; *T. canis* has one chromosome with only 2 tRNA genes [13]. Like other trichodectid lice [13], the chromosomes were compact with little non-coding sequence, and each of the chromosomes was simple, bearing only one to two protein-coding genes or one rRNA gene per chromosome along with one, two, three, or four tRNA genes (Table 2).

Natural selection has been suggested to favor compact chromosome size in lice with fragmented mitochondrial genomes based on observations that chromosomes with longer gene regions have smaller non-gene regions [12, 13]. In *G. aurei* chromosomes, we observed a strong relationship between chromosome size and gene size, with larger genes occupying larger chromosomes. However, there was not a relationship between chromosome size and the length of non-gene regions in *G. aurei*, casting some doubt on the impact of selection on chromosome size.

Whereas genes of other trichodectid lice found on the same chromosome are adjacent or very nearly so [13], that was not the case for *G. aurei* genes, which were frequently interrupted by non-gene regions between them. In other words, instead of a single non-gene region per chromosome, in *G. aurei*, there can be one or as many as six non-gene regions per chromosome (Table 2). Gaps between genes were prevalent in *G. aurei*, and more than half of the protein-coding and rRNA genes lacked a tRNA gene at the beginning and/or end of them. Similar dispersion of non-gene sequences was observed in a more distantly related bird louse, however [35]. Moreover, the tRNA-punctuation model of RNA processing [36], where each major gene begins and ends with a tRNA gene, thus allowing the transcript to be processed by cutting the protein- and rRNA-coding genes from the primary transcript at the flanking tRNAs, is not observed in *G. aurei*. While it seems to be the exception rather than the rule, some other insects, including some lice, have been shown to lack tRNA punctuation of major genes (e.g., [4, 13, 37]).

### Motifs shared among *G. aurei* chromosomes

Some portion of the non-gene region(s) of *G. aurei* mitochondrial chromosomes must serve at least two important functions, one to initiate transcription of the circle and the other to initiate replication of the chromosome itself (i.e., the D-loop; [38, 39]. Given this functional importance, we would expect to see these functionally important regions potentially shared by multiple chromosomes and relatively free of heteroplasmy. Heteroplasmy is extensive in the noncoding regions of the *G. aurei cox1* chromosome, as documented by analysis of multiple clones [17]. Heteroplasmy in non-coding regions of other *G. aurei* chromosomes was observed here in Sanger sequencing runs, necessitating cloning to read through sequences of these portions of the chromosomes. However, the 79-nt common sequence is conserved to such a degree that it provides a useful anchor for amplification primers. Pietan et al. [17] noted a highly similar sequence in a non-congeneric species of louse, *Thomomydoecus minor*, that inhabits pocket

gophers. This conservation between genera and the presence of an identical 79-nt region on every chromosome in a zone of otherwise extensive heteroplasmy suggest the possibility of evolutionary conservation and functional significance of this sequence, perhaps serving as a common promoter for the initiation of transcription. Alternatively, other authors have viewed the conservation of common sequences among different chromosomes within a species as a possible outcome of the mechanics of chromosomal recombination or gene conversion [3, 7, 8, 15].

Another region of conserved sequence, designated the CSB herein, was found roughly 100–200 nt downstream of the 79-nt region. This shared motif (Fig 2) differed slightly in sequence among chromosomes. As suggested by others, this universal element may be a result of recombination among chromosomes followed by diversification through mutation [3, 7, 8, 15]. Alternatively, this common sequence may represent a functionally significant site. The CSB sequence had a low AT content, but was surrounded on either side by regions of high AT on each chromosome, and it was preceded by a potential origin of replication site on 11 of the 12 chromosomes and was followed by a poly-thymine stretch of 9–11 nucleotides 11, 12, or 32 nucleotides away from the start of the "gene", which is consistent with this region being important in DNA replication [40]. Therefore, characteristics of the CSB could reflect chromosomal mechanics (i.e., recombination), but it could suggest the presence of functional significance, perhaps in DNA replication.

## Primitive vs. derived gene arrangements

Song et al. [13] reported an exceptionally diverse set of karyotypes among trichodectid lice. In their analysis, the only species of lice with identical mitochondrial karyotypes were species that occurred on the same or very closely related host species, consistent with cospeciation from recent common ancestors. In their character-mapping analysis, they identified eight derived mitochondrial gene cluster arrangements (protein-coding genes plus tRNA genes). All eight of these derived characters are observed in the three species of *Bovicola* that they analyzed, and they are also observed here in *G. aurei*. *Trichodectes canis* shared only six of these eight derived gene clusters. Phylogenetic analysis of the *cox1* gene indicates that *G. aurei* and *T. canis* are more closely related to one another than either is to the genus *Bovicola* (see Supplementary Material of [13]). Therefore, two of these derived gene clusters were likely lost in the lineage leading to *T. canis*.

The *G. aurei* mitochondrial genome is unusual in another respect. All eutherian mammal lice described to date have shown an *atp8-atp6* arrangement, with the two genes being adjacent to each other. As discussed by Sweet et al. [15], these genes have been understood to be co-transcribed in most species [41, 42], although two exceptions to this rule have been described in bird lice [4, 15]. The arrangement of these genes on the same chromosome but not contiguous (interrupted by 29 nt of non-coding sequence) in *G. aurei* presents yet another exception to this rule and may have implications for how mRNA processing or translation occurs in lice.

## Acknowledgments

Portions of this work were the subject of a University of Northern Iowa Master of Science thesis (A. Place). We thank Wyatt Andersen, Jillian Hill, and Dino Bolic for their assistance in the laboratory. We thank Arun Somwarpet-Seetharam and Tanya Murtha at the Iowa State University DNA Facility for their assistance. We are grateful to Alexander Donath, who provided helpful recommendations on interpretation of Mitos2 analyses, and to two anonymous reviewers.

## Author Contributions

**Conceptualization:** Theresa A. Spradling, Alexandra C. Place.

**Data curation:** Theresa A. Spradling, James W. Demastes.

**Formal analysis:** James W. Demastes.

**Investigation:** Alexandra C. Place, Ashley L. Campbell.

**Methodology:** Theresa A. Spradling, Alexandra C. Place, James W. Demastes.

**Supervision:** Theresa A. Spradling, James W. Demastes.

**Writing – original draft:** Theresa A. Spradling, James W. Demastes.

**Writing – review & editing:** Theresa A. Spradling, James W. Demastes.

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
