## [Decision Letter · Decision Letter 0]

26 Apr 2021

PONE-D-21-09156

Mitochondrial Genome of Geomydoecus aurei, a Pocket-Gopher Louse

PLOS ONE

Dear Dr. Spradling,

Thank you for submitting your manuscript to PLOS ONE. After careful consideration, we feel that it has merit but does not fully meet PLOS ONE’s publication criteria as it currently stands. Therefore, we invite you to submit a revised version of the manuscript that addresses the points raised during the review process.

Both reviewers applaud the thoroughness of this report and the contribution that your data have to the field. However, both have identified minor issues with the interpretation of some of the analyses, and these comments should be addressed in your revision.

We look forward to receiving your revised manuscript.

Kind regards,

Ross Frederick Waller, Ph.D

Academic Editor

PLOS ONE

Journal Requirements:

Reviewers' comments:

Reviewer's Responses to Questions

**Comments to the Author**

1. Is the manuscript technically sound, and do the data support the conclusions?

Reviewer #1: Yes

Reviewer #2: Yes

2. Has the statistical analysis been performed appropriately and rigorously? 

Reviewer #1: Yes

Reviewer #2: I Don't Know

3. Have the authors made all data underlying the findings in their manuscript fully available?

Reviewer #1: Yes

Reviewer #2: Yes

4. Is the manuscript presented in an intelligible fashion and written in standard English?

Reviewer #1: Yes

Reviewer #2: Yes

5. Review Comments to the Author

Reviewer #1: Spradling et al. present a very well conducted and valuable contribution to our understanding of louse mitochondrial genomics. While there are a few minor issues of interpretation and aspects in which I would prefer the authors provide a bit more detail, the paper is very close to publishable as is. Suggestions and questions are outlined below:

1) Line 56. Given that highly fragmented mitochondrial genomes are in no way synapomorphic for the clade Trichodectidae+Anoplura (Sweet 2020, 2021) and rather have evolved repeatedly across the Ischnocera, the term Mitodivisia shouldn't be used. Song et al. knew when they published that other lice had divided mt genomes (citing papers which had shown it) yet chose to ignore this in proposing such an ill fitting taxon name, we have even less reason to ignore this now that Sweet's work is coming out. That a fragmented mt genome of this type is ancestral for the clade Tricho+Ano is beyond doubt, but the clade shouldn't be named for a convergent feature.

2) Methods. The authors are to be applauded for the highly detailed description of the processes used to generate the mt genome sequences. The combination of PCR generated chromosome sequences with next generation data allows for strong comparison of assembly methods across the sequencing data types. Could you please provide some discussion on how assembly of the NGS data in Geneious (particularly through non-coding regions) compared with that of the PCR+cloning generated sequencing of the same regions?

3) Annotations. MITOS is a comparatively poor method for inferring rRNA annotations, the covarion model for such genes is much more poorly developed than the one for tRNAs (the shorter size and common cruciform arrangement makes that considerably easier). Accordingly rrnL is much larger than most other louse mt genomes sequenced to date (which are in the order or 1100-1200 bp). I suspect that it doesn't overlap H, but rather that the gene is roughly 300bp shorter than currently annotated and thus would match the minichromosome found in Damalinia (Cameron et al. 2011).

In a similar vein, I would encourage the authors to examine the gaps/overlaps identified between PCG and rRNA genes by making alignments of the MITOS inferred annotations against those of other lice to check the algorithm. MITOS is frequently deficient in determining start and stop codons when compared against other related taxa and I would not accept the annotations as outputted without further examination.

4) tRNA punctuation model. MITOS doesn't use the punctuation model in its annotation 'decisions' and has no issue with overlaps between tRNAs and PCGS - so tends to hugely underestimate incomplete stop codons. These should be revealed by doing the alignment comparisons suggested at #3, but I suspect you are misinterpreting the punctuation model - MITOS is not a good test of it's relevance to a given taxon when overlaps are inferred. Conversely, it really only applies to truncating the transcription of a gene which lacks a full stop codon, spacer sequence doesn't need to be post transcriptionally modified for the mRNA to be correctly translated (it just drops off at the stop codon), so the text at lines 303 and 315 reflect this mild misinterpretation of Ojala's model.

5) Similarity between the duplicated copies of tRNA-F should be quantified and probably also depicted in a figure (an alignment of the different chromosomal copies) such that the readers can clearly see this interesting result. Additionally, as the arrangement F-ND6 is found in all other trichodectids (and is common across Ischnocera: Cameron et al 2011), further exploration of why you consider the F-ND6-M-T-G-ND3 copy to be non-functional whereas other chromosomes are considered functional is worth exploring through such data. It's not impossible that the copy on F-ND6-M-etc has been rendered non-functional by mutations after duplication, but given that this is a strong candidate for the ancestral position in this family it bears extra scrutiny.

6) Primitive & Derived Gene Arrangements. These derived gene boundaries are somewhat misrepresented (lines 266-268) as 7 pf the 8 listed (all but E-cob-S1) are found in one or more other lice and only E-cob-S1 is reasonably synapomorphic for Tricho+Ano. This section is actually a discussion of shared derived gene boundaries in Ischnocera and should be presented as such.

Reviewer #2: This is an interesting, thorough and solid manuscript.

My only major comment is on the interpretation/annotation of trnF gene and trnF*. It is very unusual to have the same gene or its pseudo gene on all 12 minichromosomes, and they all have the opposite orientation to all other genes and all in the middle of non-coding regions. The authors will have to present enough convincing evidence to support the interpretation/annotation of trnF and trnF*. Having a secondary structure alone produced by a tRNA search program is not convincing because it can be formed simply by chance. This interpretation/annotation can be wrong and thus may mislead readers. These sequences look more like part of the non-coding regions which happen to form tRNA-like structure, based on what I can see in the current manuscript.

My minor comments are:

1. Line 33: trnF is mis-written as tRNF here and many times in the manuscript. Pls correct.

2. Line 63: correct tRNI to trnI.

3. Line 198: Table 2, row 3: pls check rrnL annotation. 1410 nt seems too long thus overlapping with trnV could be over-estimated. rrnL is ~1100 nt in some lice reported.

4. Line 198: Table 2, row 4: atp6/8 should be atp8/6.

5. Line 198: Table 2, row 5: Cob should be cob.

6. Table 2 bottom note: correct tRNAF to trnF.

7. Line 223: verify "six tRNA genes overlapping other genes by 7-64 nucleotides" after checking gene annotation.

8. Line 235: tRNAF to trnF.

9. Line 247: atp6/atp8 should be atp8/atp6.

10. Line 254: briefly explain how MITOS [30] works to identify a putative origin of replication in mt minichromosomes.

11. Line 267: nat1 should be nad1.

12. Figure 1: pls check rrnL annotation. 1410 nt seems too long thus overlapping with trnV could be over-estimated.

13. Figure 1: atp6 should be atp8, and vice versa.

14. Figure 1: pls check nad2 and nad4 annotation. Is there is an alternative annotation of their start site?

6. PLOS authors have the option to publish the peer review history of their article (what does this mean?). If published, this will include your full peer review and any attached files.

Reviewer #1: No

Reviewer #2: No

---

## [Author Response · Author response to Decision Letter 0]

31 May 2021

Reviewer #1: Spradling et al. present a very well conducted and valuable contribution to our understanding of louse mitochondrial genomics. While there are a few minor issues of interpretation and aspects in which I would prefer the authors provide a bit more detail, the paper is very close to publishable as is. Suggestions and questions are outlined below:

1) Line 56. Given that highly fragmented mitochondrial genomes are in no way synapomorphic for the clade Trichodectidae+Anoplura (Sweet 2020, 2021) and rather have evolved repeatedly across the Ischnocera, the term Mitodivisia shouldn't be used. Song et al. knew when they published that other lice had divided mt genomes (citing papers which had shown it) yet chose to ignore this in proposing such an ill fitting taxon name, we have even less reason to ignore this now that Sweet's work is coming out. That a fragmented mt genome of this type is ancestral for the clade Tricho+Ano is beyond doubt, but the clade shouldn't be named for a convergent feature. 

***We agree. Taxonomic use removed from three places in manuscript.

2) Methods. The authors are to be applauded for the highly detailed description of the processes used to generate the mt genome sequences. The combination of PCR generated chromosome sequences with next generation data allows for strong comparison of assembly methods across the sequencing data types. Could you please provide some discussion on how assembly of the NGS data in Geneious (particularly through non-coding regions) compared with that of the PCR+cloning generated sequencing of the same regions? 

***See new paragraph at end of “Genome Analysis” section in Methods.

3) Annotations. MITOS is a comparatively poor method for inferring rRNA annotations, the covarion model for such genes is much more poorly developed than the one for tRNAs (the shorter size and common cruciform arrangement makes that considerably easier). Accordingly rrnL is much larger than most other louse mt genomes sequenced to date (which are in the order or 1100-1200 bp). I suspect that it doesn't overlap H, but rather that the gene is roughly 300bp shorter than currently annotated and thus would match the minichromosome found in Damalinia (Cameron et al. 2011). 

***On reinspection (using Mitos2 but limiting the search to the single feature of interest), we find the rrnL gene to be smaller. We reached out to Alexander Donath for support with Mitos2, and determined that it is most likely that the annotation software is “tricked” into returning a longer length of rrnL that includes the surrounding tRNAs. A similar problem seen when T. canis sequence is run through Mitos2. Sequence alignment is difficult, but we examined alignments both in Geneious and using Beagle. In the end, we trimmed the rrnL gene annotation to remove the overlap with trnH and trnV. We removed text about overlapping genes as our evidence for this is poor.

In a similar vein, I would encourage the authors to examine the gaps/overlaps identified between PCG and rRNA genes by making alignments of the MITOS inferred annotations against those of other lice to check the algorithm. MITOS is frequently deficient in determining start and stop codons when compared against other related taxa and I would not accept the annotations as outputted without further examination.

***PCG boundaries have been reevaluated, particularly in cases where we see potential overlap with tRNA genes. Using MITOS2 again, but limiting the search to the single feature of interest, we’ve identified new boundaries for nad1, nad2, and nad6 that we think are more consistent with what has been reported for other species. This reanalysis removes the overlap between nad2 and trnN. The additional slight overlap in PCG and tRNA boundaries still exists for nad1, nad4, cox1 (Pietan et al.), specifically at the beginning of nad4, nad1, and cox1 (Pietan et al.) genes where there are multiple potential start sites, but manual alignment of these gene sequences with other trichodectid genes is not a straightforward process in these regions given the amount of sequence divergence at those sites. We trimmed the annotation to begin at the next available start site that did not overlap with the upstream tRNA gene, and we edited the text and figure accordingly, also partly as a result of our conversation with Alexander Donath. 

4) tRNA punctuation model. MITOS doesn't use the punctuation model in its annotation 'decisions' and has no issue with overlaps between tRNAs and PCGS - so tends to hugely underestimate incomplete stop codons. These should be revealed by doing the alignment comparisons suggested at #3, but I suspect you are misinterpreting the punctuation model - MITOS is not a good test of it's relevance to a given taxon when overlaps are inferred. Conversely, it really only applies to truncating the transcription of a gene which lacks a full stop codon, spacer sequence doesn't need to be post transcriptionally modified for the mRNA to be correctly translated (it just drops off at the stop codon), so the text at lines 303 and 315 reflect this mild misinterpretation of Ojala's model.

***All of our sequences have full stop codons. The text the tRNA punctuation model has been edited in last paragraph of “Chromosome Architecture” in Discussion.

5) Similarity between the duplicated copies of tRNA-F should be quantified and probably also depicted in a figure (an alignment of the different chromosomal copies) such that the readers can clearly see this interesting result. Additionally, as the arrangement F-ND6 is found in all other trichodectids (and is common across Ischnocera: Cameron et al 2011), further exploration of why you consider the F-ND6-M-T-G-ND3 copy to be non-functional whereas other chromosomes are considered functional is worth exploring through such data. It's not impossible that the copy on F-ND6-M-etc has been rendered non-functional by mutations after duplication, but given that this is a strong candidate for the ancestral position in this family it bears extra scrutiny. 

***Fig. 2 has been added to provide an alignment. Now that we have reevaluated the ND3 chromosome, we feel confident that there is indeed a functional trnF gene on that chromosome, near the ancestral position (near ND6). We have reframed our perspective of the sequences we were calling trnF or trnF-like, and now refer to these as a conserved sequence block (CSB), consistent with other authors.

 

6) Primitive & Derived Gene Arrangements. These derived gene boundaries are somewhat misrepresented (lines 266-268) as 7 pf the 8 listed (all but E-cob-S1) are found in one or more other lice and only E-cob-S1 is reasonably synapomorphic for Tricho+Ano. This section is actually a discussion of shared derived gene boundaries in Ischnocera and should be presented as such. 

***We have clarified this (first sentence of the section titled, “Primitive vs. Derived Gene Arrangements” in the Results.

Reviewer #2: This is an interesting, thorough and solid manuscript.

My only major comment is on the interpretation/annotation of trnF gene and trnF*. It is very unusual to have the same gene or its pseudo gene on all 12 minichromosomes, and they all have the opposite orientation to all other genes and all in the middle of non-coding regions. The authors will have to present enough convincing evidence to support the interpretation/annotation of trnF and trnF*. Having a secondary structure alone produced by a tRNA search program is not convincing because it can be formed simply by chance. This interpretation/annotation can be wrong and thus may mislead readers. These sequences look more like part of the non-coding regions which happen to form tRNA-like structure, based on what I can see in the current manuscript.

***On further analysis and inspection of the nad3/nad6 chromosome, we see that we were missing the functional copy of trnF in a position that would be expected based on sequences of other trichodectid lice. We have submitted a new Genbank entry for this chromosome (and also for nad2 and 16s). All statistical analyses that used gene size have been repeated and new numbers inserted in the text. The sequence we previously referred to as trnF or trnF like is now called CSB (conserved sequence block) and Figure 2 has been created to show that alignment.

My minor comments are:

1. Line 33: trnF is mis-written as tRNF here and many times in the manuscript. Pls correct. ***Completed.

2. Line 63: correct tRNI to trnI. ***Corrected.

3. Line 198: Table 2, row 3: pls check rrnL annotation. 1410 nt seems too long thus overlapping with trnV could be over-estimated. rrnL is ~1100 nt in some lice reported. 

***On reinspection, we find the rrnL gene to be 1,058 bp, which seems more reasonable in comparison to other species, as you point out. 

4. Line 198: Table 2, row 4: atp6/8 should be atp8/6. ***Corrected

5. Line 198: Table 2, row 5: Cob should be cob. ***Corrected

6. Table 2 bottom note: correct tRNAF to trnF. ***Corrected

7. Line 223: verify "six tRNA genes overlapping other genes by 7-64 nucleotides" after checking gene annotation. 

***As described in our response to reviewer 1, we have reanalyzed data, altered some gene annotations, and removed discussion of overlapping genes.

8. Line 235: tRNAF to trnF. ***Corrected

9. Line 247: atp6/atp8 should be atp8/atp6. ***Corrected

10. Line 254: briefly explain how MITOS [30] works to identify a putative origin of replication in mt minichromosomes. 

***This has been added to the Methods section, first paragraph of “Annotation and Analysis of Genome Size.”

11. Line 267: nat1 should be nad1. ***Corrected

12. Figure 1: pls check rrnL annotation. 1410 nt seems too long thus overlapping with trnV could be over-estimated. 

***On reanalysis, we find the rrnL gene to be 1,058 nt, placing the gene size just smaller than that of Trichodectes canis (1,092 nt). The new gene boundary annotations have been submitted to Genbank.

13. Figure 1: atp6 should be atp8, and vice versa. Corrected

14. Figure 1: pls check nad2 and nad4 annotation. Is there is an alternative annotation of their start site? ***The nad2 chromosome has been reinspected. We adjusted the placement of the start site. It aligns well with the Trichodectes canis nad2 start and produces a gene that is 981 nt, which is within 42 bp of the T. canis gene length. The nad4 chromosome and gene boundaries have been reinspected. We believe the start and end sites are both correct, and they produce a gene length of 1,338 nt, which is within 21 bp of the gene length in T. canis.

---

## [Editor Report · Decision Letter 1]

21 Jun 2021

Mitochondrial Genome of Geomydoecus aurei, a Pocket-Gopher Louse

PONE-D-21-09156R1

Dear Dr. Spradling,

We’re pleased to inform you that your manuscript has been judged scientifically suitable for publication and will be formally accepted for publication once it meets all outstanding technical requirements.

Kind regards,

Ross Frederick Waller, Ph.D

Academic Editor

PLOS ONE
---

## [Editor Report · Acceptance letter]

19 Jul 2021

PONE-D-21-09156R1 

Mitochondrial Genome of *Geomydoecus aurei*, a Pocket-Gopher Louse 

Dear Dr. Spradling:

I'm pleased to inform you that your manuscript has been deemed suitable for publication in PLOS ONE. Congratulations! Your manuscript is now with our production department. 

Kind regards, 

on behalf of

Dr. Ross Frederick Waller 

Academic Editor

PLOS ONE